# Expanding Pharmacotherapy Data Collection, Analysis, and Implementation in ERAS^®^ Programs—The Methodology of an Exploratory Feasibility Study

**DOI:** 10.3390/healthcare8030252

**Published:** 2020-08-03

**Authors:** Eric Johnson, Richard Parrish, Gregg Nelson, Kevin Elias, Brian Kramer, Marian Gaviola

**Affiliations:** 1Department of Pharmacy Services, University of Kentucky, Lexington, KY 40506, USA; eric.johnson@uky.edu; 2Department of Biomedical Sciences, Mercer University School of Medicine, Macon, GA 31207, USA; 3Department of Obstetrics and Gynecology, University of Calgary, Calgary, AB T2N 1N4, Canada; Gregg.Nelson@albertahealthservices.ca; 4Department of Obstetrics, Gynecology and Reproductive Biology, Harvard Medical School, Boston, MA 02115, USA; kelias@bwh.harvard.edu; 5Department of Pharmacy Services, Grant Medical Center, Columbus, OH 43215, USA; bkramer321@gmail.com; 6College of Pharmacy, University of North Texas, Denton, TX 76203, USA; marian.gaviola@gmail.com

**Keywords:** collaboration, enhanced recovery, infection, surgical wound, perioperative care, pharmacy, clinical, post-operative nausea and vomiting, prophylaxis, surgeon, surgery, colorectal, surgery, gynecological, thromboembolism, venous

## Abstract

Surgical organizations dedicated to the improvement of patient outcomes have led to a worldwide paradigm shift in perioperative patient care. Since 2012, the Enhanced Recovery After Surgery (ERAS^®^) Society has published guidelines pertaining to perioperative care in numerous disciplines including elective colorectal and gynecologic/oncology surgery patients. The ERAS^®^ and ERAS-USA^®^ Societies use standardized methodology for collecting and assessing various surgical parameters in real-time during the operative process. These multi-disciplinary groups have constructed a bundled framework of perioperative care that entails 22 specific components of clinical interventions, which are logged in a central database, allowing a system of audit and feedback. Of these 22 recommendations, nine of them specifically involve the use of medications or pharmacotherapy. This retrospective comparative pharmacotherapy project will address the potential need to (1) collect more specific pharmacotherapy data within the existing ERAS Interactive Audit System^®^ (EIAS) program, (2) understand the relationship between medication regimen and patient outcomes, and (3) minimize variability in pharmacotherapy use in the elective colorectal and gynecologic/oncology surgical cohort. Primary outcomes measures include data related to surgical site infections, venous thromboembolism, and post-operative nausea and vomiting as well as patient satisfaction, the frequency and severity of post-operative complications, length of stay, and hospital re-admission at 7 and 30 days, respectively. The methodology of this collaborative research project is described.

## 1. Introduction

Since 2012, the Enhanced Recovery After Surgery (ERAS^®^) Society has published guidelines pertaining to perioperative care in numerous disciplines including elective colorectal [1] and gynecologic/oncology [2,3,4] surgery patients. These bundled guidelines contain recommendations on the use of pharmacologic therapy, including prophylaxis for (1) surgical site infection (SSI), (2) thromboembolism (VTE), and (3) postoperative nausea and vomiting (PONV), among others. While the guidelines contain high-quality evidence of the use of pharmacotherapy in each of these ERAS program elements, the specifics of agent selection and dosing regimens are absent. These dosing variables include medication administration time in relation to the procedure, dose of medication used, and duration of therapy. The literature suggests that the lack of effective prophylaxis to address these three endpoints is associated with significant clinical morbidity, and they may be independent drivers of hospital length of stay. Suboptimal preventive pharmacotherapy may lead to increased complication rates and delayed patient discharge from the facility.

The ERAS^®^ and ERAS-USA^®^ Societies use a standardized methodology for collecting and assessing various surgical parameters in real-time during the operative process [5]. By utilizing a retrospective multi-center research design, this project will address the potential need to (1) collect more specific pharmacotherapy data within the existing ERAS Interactive Audit System^®^ (EIAS) program, (2) understand the relationship between medication regimen and patient outcomes, and (3) minimize variability in pharmacotherapy use in the elective colorectal and gynecologic/oncology surgical cohort. The specific aims of this project include:Creation of a pharmacotherapy database and execution of a retrospective analysis to compile perioperative medication-specific data related to significant improvements in patient outcomes.Estimation of the impact of prophylaxis medications on length of stay, postoperative complications, and hospital readmission rates at 7 and 30 days for the following indications:Surgical site infections;Thromboembolism; Post-operative nausea and vomiting.Provide guidance on optimal medication use regarding regimen selection, dosing, timing, and duration of therapy.

## 2. Research Strategy

The development and evolution of Enhanced Recovery Programs have led to significant improvements in the care of surgical patients, as well as a decrease in important benchmarks such as hospital length of stay (LOS) and postoperative complications [6]. As a result, surgical organizations dedicated to the improvement of patient outcomes have led a paradigm shift in perioperative patient care. Specific groups, like the Enhanced Recovery After Surgery (ERAS^®^) Society and ERAS^®^ USA, have constructed a bundled framework of care entailing 22 specific components of perioperative clinical interventions, which are logged in a central database, allowing a system of audit and feedback. Of these 22 recommendations, nine of them specifically involve the use of medications or pharmacotherapy. They include the following: (1) pre-anesthetic medication; (2) prophylaxis against venous thromboembolism (VTE); (3) antimicrobial prophylaxis and skin preparation; (4) standard anesthetic protocol; (5) post-operative nausea and vomiting (PONV) prophylaxis; (6) perioperative fluid management; (7) prevention of postoperative ileus (including use of postoperative laxatives); (8) postoperative analgesia, and (9) postoperative glucose control [7]. While these recommendations address global concepts of perioperative patient care, the ERAS protocols do not specify particular pharmacotherapeutic medication classes, agents, or doses. As a result of the inherent variability in medication use, the optimal pharmacotherapeutic agents within ERAS^®^ pathways are unknown. Furthermore, variance in the timing of medication administration leaves practitioners searching for the exact method of replicating the significant outcomes found in ERAS publications.

In its current form, EIAS^®^ collects limited information related to medication administration for ERAS^®^ patients. Despite this dearth, patient outcomes have consistently improved in institutions that have adopted ERAS^®^ pathways. Whether these improvements are due to individual therapeutic agents or the application of a bundled approach is unknown. Practitioners and pharmacists are challenged to make evidence-based pharmacotherapeutic recommendations of agents within the protocol. Inevitably, debates on implementation often center on more costly versions of medications such as intravenous acetaminophen or liposomal bupivacaine as means to limit opioid use.

We plan to integrate de-identified patient data from two separate ERAS^®^ centers in North America with pharmacotherapy data collected retrospectively from each site. From this registry, we will seek answers to comparative pharmacotherapy questions embedded in the ERAS^®^ pathway. Specifically, we plan to evaluate the following: (1) timing of preoperative and post-operative thromboprophylaxis and the impact on post-operative VTE; (2) specific agents and doses of antimicrobials used in surgical antimicrobial prophylaxis, and (3) optimal and efficacious regimens in the successful prevention of PONV.

## 3. Approach

Our Enhanced Recovery Comparative Pharmacotherapy Collaborative (ERCPC) group plans to evaluate the role that specific pharmacotherapeutic regimens within the ERAS^®^ protocol play in regard to the improved outcomes, readmission, and hospital LOS. In addition, data regarding patient experience or satisfaction scores will be collected. This information will be obtained through patient registry of the EIAS^®^. The two institutions that have provided written support for access to their patients’ data are the Foothills Medical Centre (FMC) of Calgary, Alberta, Canada and the Brigham & Women’s Hospital of Boston, MA, USA. Both institutions have robust ERAS^®^ practices and are leading researchers in the practice of enhanced surgical recovery.

Data from patient healthcare records at each site will be collected and entered into a centralized REDCap database. The project’s data dictionary is included in a Appendix A attachment. Patient demographics, intraoperative anesthetic techniques, and procedure details will also be collected. Drug-related variables will be compared to determine the effect of agent use on outcome measures. Potential hurdles that we anticipate are low event rates with some primary outcomes measures, specifically VTE. Recent literature suggests that the incidence of VTE in colorectal surgery patients is approximately 2.2% [8,9]. While a population of 500 colorectal patients would have an estimated incidence of 11 cases, we may be challenged to obtain a difference between groups if numerous different regimens are used. Gynecologic and colorectal malignancy patients show a similarly low incidence; however, it is slightly higher at approximately 3% [10] and is purported to be on the rise [11]. By combining the two patient populations, we estimate a sufficient number of thromboembolic events from which we will be able to ascertain a statistical difference. Additionally, because we will be evaluating the pharmacotherapeutic interventions from ERAS^®^, it is possible that ERAS^®^ components not captured in our analysis may play more significant roles in reducing negative outcomes compared to the agents that we evaluate. However, if no difference is found, this may too provide justification for the use of different regimens within the ERAS pathway. Finally, we have strong physician support from experienced researchers who are eager to participate in this project.

## 4. Specific Research Questions

Determine the *optimal antimicrobial agents used in surgical prophylaxis*, including pre-operative dose, timing of preoperative dose, intraoperative repeat doses, postoperative duration of therapy, classification of surgical site infection (if present), and infection organism (if applicable) [12].Provide evidence to define *optimal prophylaxis regimens to prevent PONV* in this surgical population. Specific parameters of analysis include PONV risk factors, preoperative Apfel risk score [13], prophylaxis regimen (dose, timing), postoperative nausea, and duration of Post Anesthesia Care Unit (PACU) LOS [14].Evaluate the effect that *venous thromboembolism (VTE) prophylaxis* provides in preventing post-operative VTE in high-risk oncology populations. Specific points of evaluation include prophylaxis agent used (unfractionated heparin versus low-molecular weight heparin verses direct thrombin inhibitors), perioperative timing of dose, post-operative duration of therapy, thromboembolic risk factors, and patient weight [15,16].

## 5. Conclusions

Our ERCPC group plans to evaluate the impact that specific pharmacotherapeutic regimens within the ERAS^®^ protocol have on primary clinical outcomes (surgical site infections, venous thromboembolism, and post-operative nausea and vomiting) as well as their relationship to and impact on readmission, complications, and hospital LOS. We plan to integrate de-identified patient data from two separate ERAS^®^ centers in North America with pharmacotherapy data collected retrospectively from each site. From this registry, we will seek answers to comparative pharmacotherapy questions embedded in the ERAS^®^ pathway. Specifically, we plan to evaluate the following: (1) specific agents and doses of antimicrobials used in surgical antimicrobial prophylaxis; (2) timing of preoperative and post-operative thromboprophylaxis and the impact on post-operative VTE, and (3) optimal and efficacious regimens in the successful prevention of PONV.

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
