# Peer review of "Expanding Pharmacotherapy Data Collection, Analysis, and Implementation in ERAS® Programs—The Methodology of an Exploratory Feasibility Study"

_healthcare, 2020, doi:10.3390/healthcare8030252_

Round 1

Reviewer 1 Report

The authors should be commended for bringing to the attention of the personnel involved in the care of surgical patients the issue of postoperative pharmacotherapy. There is lack of evidence in this area regarding drug dosing and timing of administration. Based on the above considerations, a feasibility study is much needed. The manuscript is clear and well-written.

Author Response

We thank the reviewer for these kind comments and observations.

Reviewer 2 Report

This is a methods paper that seeks to identify how to go about learning which are the most ideal medications for prophylaxis to use when discussing ERAS protocols. 

There have been multiple studies looking at each of these topics in the past (PONV prophylaxis, antimicrobial agents, and prevention of VTE). The novelty of this proposed study is looking at all three together in the context of enhanced recovery after surgery. 

Though these questions may play a role in ERAS, the question is how much of a role do they play. This type of a study may help identify this. 

Author Response

We appreciate the reviewer's insights, comments, and encouragement.

Reviewer 3 Report

The authors presented an important topic and in the introduction tried to explain the scientific basis of the conducted research. Literature should be ordered according to guidelines.

The results of surgical treatment are closely related to the overall procedure package, which covers all aspects of perioperative treatment. It is a team and multidisciplinary proceeding. One of it's elements is pharmacotherapy. The authors presented a very interesting study design that will allow for the selection, modification and systematization of perioperative pharmacotherapy. Consequently, it will shorten the patient's stay in hospital and reduce the risk of complications associated with surgical treatment. The project presented for assessment has a considerable cognitive value and its foundation is numerous up to date scientific evidence. Methodologically prepared without reservations. I only have a remark to the way citations and the list of references were introduced. I recommend that the authors of the thesis introduce editorial corrections regarding the literature (following the publisher's guidelines).

Author Response

We appreciate the reviewer's insights and recommendations to introduce literature in accordance with the publisher's guidelines. We have corrected the reference style by numbering the citations consecutively in the manuscript and reference section.

Reviewer 4 Report

This is a proposed project, without data collection or conclusions.  As a report, seems to be an interesting and useful proposal.

Some minor edits:

Line 8:  typo in 20122012

Line 32:  Correct 2013; 2019

Line 33:  prophylaxis for (not against).

Line 59:  Development and evolution (not maturation).

Line 78:  limitedlimited

Author Response

We appreciate the reviewer's careful and supportive comments and suggestions. 

Reviewer: Line 8:  typo in 20122012

Authors: we have corrected it to say '2012'

Reviewer: Line 32:  Correct 2013; 2019

Authors: we corrected the reference to 2019.

Reviewer: Line 33:  prophylaxis for (not against).

Authors: we corrected the word choice

Reviewer: Line 59:  Development and evolution (not maturation).

Authors: we corrected the word choice.

Reviewer: Line 78:  limitedlimited

Authors: we have corrected the duplicate word.